# Flax Biomass Conversion via Controlled Oxidation: Facile Tuning of Physicochemical Properties

**DOI:** 10.3390/bioengineering7020038

**Published:** 2020-04-27

**Authors:** Leila Dehabadi, Abdalla H. Karoyo, Majid Soleimani, Wahab O. Alabi, Carey J. Simonson, Lee D. Wilson

**Affiliations:** 1Department of Chemistry, University of Saskatchewan, 110 Science Place, Saskatoon, SK S7N 5C9, Canada; 2Dr. Ma’s Laboratories Inc., Unit 4, 8118 North Fraser Way, Burnaby, BC V5J 0E5, Canada; 3Department of Chemical and Biological Engineering, University of Saskatchewan, 57 Campus Drive, Saskatoon, SK S7N 5A9, Canada; 4Department of Mechanical Engineering, University of Saskatchewan, 57 Campus Drive, Saskatoon, SK S7N 5A9, Canada

**Keywords:** linen fibers, physicochemical properties, adsorption, swelling, hydration, chemical treatment

## Abstract

The role of chemical modification of pristine linen fiber (LF) on its physicochemical and adsorption properties is reported in this contribution. The surface and textural properties of the pristine LF and its peroxyacetic acid- (PAF) and chlorite-treated (CF) fiber forms were characterized by several complementary methods: spectroscopy (SEM, TEM, FT-IR, and XPS), thermal analysis (DSC and TGA), gas/water adsorption isotherms, and zeta potential (ξ). The results obtained reveal that the surface charge and textural properties (surface area and pore structure) of the LF material was modified upon chemical treatment, as indicated by changes in the biomass composition, morphology, ξ-values, and water/dye uptake properties of the fiber samples. Particularly, the pristine LF sample displays preferential removal efficiency (E_R_) of methylene blue (MB) dye with E_R_ ~3-fold greater (E_R_~62%) as compared to the modified materials (CF or PAF; E_R_~21%), due to the role of surface charge of pectins and lignins present in pristine LF. At higher MB concentration, the relative E_R_ values for LF (~19%) relative to CF or PAF (~16%) reveal the greater role of micropore adsorption sites due to the contributing effect of the textural porosity observed for the modified flax biomass at these conditions. Similar trends occur for the adsorption of water in the liquid vs. vapour phases. The chemical treatment of LF alters the polarity/charge of the surface functional groups, and pore structure properties of the chemically treated fibers, according to the variable hydration properties. The surface and textural properties of LF are altered upon chemical modification, according to the variable adsorption properties with liquid water (*l*) vs. water vapor (*g*) due to the role of surface- vs. pore-sites. This study contributes to an understanding of the *structure-adsorption* properties for pristine and oxidized flax fiber biomass. The chemical conversion of such biomass yields biomaterials with tunable surface and textural properties, as evidenced by the unique adsorption properties observed for pristine LF and its modified forms (CF and PAF). This study addresses knowledge gaps in the field by contributing insight on the relationship between *structure and adsorption* properties of such LF biomass in its pristine and chemically modified forms.

## 1. Introduction

Nowadays, eco-friendly and renewable materials have attracted increasing attention in the quest to address environmental sustainability and concerns over energy production [1]. Relative to cotton fibers, flax (*Linum usitatissimum*) fiber derived from the linseed plant is one of the most common and naturally occurring fibers. Raw flax is normally processed by a method known as retting, where the fibers are separated from the woody core cells and cuticular epidermis to generate clean fibers devoid of surface-bound residues [2,3,4]. Flax fibers exemplify a category of biomass composites with variable biopolymer composition, where cellulose, hemicellulose, lignins and pectins are the primary constituents [5]. The presence of cellulose microfibrils give flax its crystalline nature [6], where abundant hydroxyl (–OH) groups and their surface accessibility influence the hydrophile-lipophile balance (HLB) of the biomaterial. In contrast, hemicellulose polysaccharides which comprise the major fraction of the branched and amorphous parts of the fiber are known to play a key role in water absorption [7]. Similarly, pectins play a key role in water absorption due to its functional role as a binder within the composite structure of flax fiber [8,9]. Pectins are hydrophilic polysaccharides composed of α-1,4 linked D-galacturonic acids that possess a complex and branched structure [4]. By contrast, lignins are polyphenolic constituents of flax that have a complex structure with a relatively apolar character [10]. Recently, the use of natural flax fibers as reinforcement or as a filler agent in bio-composites has increased drastically owing to the low density, high stiffness and biodegradable nature of such biomass [11]. However, the hygroscopic nature of these materials presents a challenge due to the swelling of the fibers that contribute to crack formation of the composites and subsequent degradation of their mechanical properties. By contrast, the use of flax fibers as bio-sorbents is well-established, especially in fields related to environmental remediation of water and wastewater [5,12]. Biomass valorization through physical and/or chemical modification yields materials with tunable textural and/or surface properties for tailored applications.

The chemical composition, surface chemistry and the mechanical strength of biomass materials have been modified by facile physical methods such as grinding/ball milling and sieving [13]. By contrast, benzoylation, acetylation, silane and peroxide treatment are some methods used to alter the physicochemical properties of bio-fibers [12,14,15]. Studies of bio-composite formation that employ chemically modified flax fiber materials with a range of polymer matrices are known [11,16]. Recent efforts are focused on developing an improved understanding of the role of changes on the physico-chemical and -mechanical properties upon chemical modification of linen fibers (LFs) [16]. In general, the surface and textural properties of biomass (e.g., starches and flax fibers) play a significant role in the uptake properties of water in the liquid (*l*) and vapour (*g*) states. Despite the high natural abundance, low cost, and high mechanical strength of cellulosic fibers, their limited utility in chemical separations, energy recovery applications and as filler agents in bio-composite formation often relates to limitations in processing. The factors that limit the widespread utilization of LFs include the following: (1) the hydrophilic nature of flax accounts for its poor mechanical and thermal stability, and (2) the limited understanding of the *structure-function* relationship of its pristine and/or modified LF biomass reveals a key knowledge gap in the field. It is posited that controlled chemical modification of LF biomass will alter the surface chemistry and textural properties in a controlled manner to afford biodegradable materials with incremental changes to their *structure-function* properties. Specifically, the adsorption properties of modified biomaterials show tunable water uptake due to incremental variation of the surface chemical and textural properties of LFs upon chemical modification.

To overcome the limitation of flax fibers along with the knowledge gap concerning the *structure-function* properties identified above, this study examines the *structure-adsorption* properties of pristine flax fiber (LF) and its chemically modified forms. Modified LFs were obtained upon treatment with peroxyacetic acid and chlorite, hereafter referred to as PAF and CF, respectively. The physicochemical properties of LF and its chemically modified forms (PAF and CF) were characterized via several complementary methods: spectroscopy (Fourier-transform infrared and X-ray photoelectron), thermal analysis (differential scanning calorimetry, thermogravimetric analysis), and surface charge (zeta-potential) methods. Gas adsorption isotherms were obtained using nitrogen and water vapour to study the uptake properties of the pristine and modified biomass, along with sorption of liquid water and aqueous methylene blue dye to study the textural properties of such biomaterials. Herein, the following objectives are addressed: (1) To convert LF to chemically modified forms using a facile chemical synthesis strategy; (2) To characterize the physicochemical properties (e.g., composition, surface chemistry, textural properties and thermal stability) of pristine and modified LFs; and (3) To study the adsorption properties of pristine and modified LFs with water (vapour and liquid). The results of this study will address knowledge gaps in the field by contributing insight on the relationship between the physicochemical properties of pristine and modified biomaterials. This study on the controlled conversion of biomass is envisaged to contribute positively to the field of advanced biomaterials to yield valorized products with tunable surface properties (chemical functionality and textural) for tailored materials in diverse applications such as environmental remediation, energy recovery, and bio-composites with improved physicochemical properties [17].

## 2. Materials and Methods

Linen fibers (LF; short and retted fiber) with a density of 1.52 g/cm^3^ was provided by Biolin Research Inc. (Saskatoon, SK, Canada). Chlorite, peroxyacetic acid and FT-IR spectroscopic-grade potassium bromide (KBr) were obtained from Sigma Aldrich (Oakville, ON, Canada) and used as received. All aqueous solutions were prepared using deionized water.

### 2.1. Pretreatment of Raw Linen Fiber

The method used to prepare the chemically modified LF materials was adapted from the literature [18]. In brief, the chlorite-treated linen fiber (CF) preparation is described, where 5 g of LF was transferred into a large vessel containing 3 g each of chlorite and acetic acid in 100 mL of water. The aqueous content was refluxed for ~8 h, followed by filtration (Whatman #2) of the mixture to obtain a solid product. The product was washed with deionized water several times to remove unreacted chlorite followed by drying in an oven at 70 °C for 48 h. In the case of peroxyacetic acid-treated (PAF) preparation, 5 g of LF fiber were transferred into a round-bottom flask containing ca. 27 mL peroxyacetic acid in water. The mixture was stirred for 48 h at 25 °C, followed by filtration and washing of the solid product with deionized water until a neutral pH was reached in the filtrate, followed by oven drying at 70 °C for 48 h.

### 2.2. Characterization Methods of Raw Linen Fiber and Its Treated Forms

#### 2.2.1. Compositional Analysis

The linen fiber composition was determined by an AOAC standard method 942.05 (2000) [19]. The lignin content (L), Acid Detergent Fiber (ADF) and Neutral Detergent Fiber (NDF) were determined as per ANKOM method 08/05 (2005), ANKOM Method 5 (2006a) and ANKOM Method 6 (2006b) on a dry matter basis [20]. Cellulose and hemicellulose contents were calculated as ADF–L and NDF–ADF, respectively [21].

#### 2.2.2. FT-IR, TGA, Nitrogen Gas Adsorption, SEM and TEM

FT-IR spectra were acquired using an FTS-40 spectrometer (Bio-Rad, Mississauga, ON, Canada) in reflectance mode against a background KBr spectrum. Powdered samples (~5 mg) were mixed with ca. 80 wt.% spectroscopic-grade KBr using a mortar and pestle followed by drying at 60 °C. The diffuse reflectance infrared Fourier transform (DRIFT) spectra for the powdered samples were obtained at 23 °C with a spectral resolution of 4 cm^−1^ over a fixed spectral (400–4000 cm^−1^) region.

The thermal stability of the LF sample and its treated forms was assessed using a TGA Q50 (TA Instruments, New Castle, DE, USA) thermal analyzer. The solid samples (20 ± 0.2 mg) were analyzed in aluminum pans with a heating rate of 5 °C·min^−1^ from 30 to 500 °C, under a N_2_ purge gas. The integration of the TGA peaks was carried out using the commercial TA Q50 software.

The surface area (SA) and pore structure (textural) properties of the LF sample and its treated forms (PAF and CF) were measured using nitrogen gas adsorption using an ASAP 2020 (Micromeritics, Norcross, GA, USA) analyzer with an accuracy of ±5%. The instrument was calibrated using a standard alumina sample (Micromeritics) with a known pore volume (PV) and SA properties. Prior to analysis, the test samples (~1 mg) were degassed at an evacuation rate of 5 mmHg·s^−1^ at 100 °C for 48 h to a stable outgas rate of <10 mmHg·min^−1^ in the sample chamber. The micropore SA was obtained using a *t*-plot (de Boer method) [22]. The Barret–Joyner–Halenda (BJH) method was employed to estimate the pore diameter (PD), where slit-shaped pores were assumed [22,23].

Scanning Electron Microscopy (SEM) images were obtained using FEG-SEM SU6600 instruments (Hitachi High Tech Co., Toronto, ON, Canada) at an accelerating voltage of 15 kV.

Transmission Electron Microscopy (TEM) was performed by negative staining of samples. A drop of liposomal sample was placed on copper-formvar coated TEM grid that was allowed to settle onto the grid surface for 1 min. Excess liquid was removed using an absorbent tissue. Staining of the grid was carried out using 0.5% phosphotungstic acid for 30 s, where the excess stain was removed. Imaging was carried out using a HT 7700 TEM device (Hitachi, city, Japan) at 80 kV.

#### 2.2.3. Raman Spectroscopy

The samples for Raman spectroscopy were prepared by imbibing the solid samples in a 10 wt.% D_2_O in H_2_O binary solvent system. The LF sample and its treated forms were isolated by centrifugation (Precision Micro-Semi Micro Centricone, Precision Scientific Co., Delhi, India) at 1800 rpm for 1 h, followed by immediate spectral analysis to minimize moisture loss. The Raman spectra were acquired using red laser (λ = 785 nm) at the following operating conditions: scan range (3500−500 cm^−1^), 10 mW laser power with 100% load, magnification (50×), cosmic ray removal, 30 s detection time, and multiple (*n* = 15) accumulative scans. All spectra were normalized and baseline corrected.

#### 2.2.4. Particle Size Distribution and Zeta Potential

The particle size distribution (PSD) and zeta potential (ξ) values were measured using a Malvern Zetasizer Nano ZS instrument (Malvern Instruments Ltd., Malvern, Worcestershire, UK). A fixed amount of sample (ca. 40 mg) was suspended in 7 mL Millipore water in 4-dram vials, where aliquots of the suspension were used to measure ξ -values at constant pH value (pH ~ 6.5).

The sample size distribution was obtained by measurement of scattered light (θ = 173°) by particles (dynamic light scattering, DLS) illuminated with a laser beam. The CONTIN algorithm was used to analyze the decay rates as a function of the translational diffusion coefficient (D) of the particles. The ξ -values and PSD were estimated from triplicate determinations, each consisting of a minimum of ten individual runs giving an estimated uncertainty of ±5%.

#### 2.2.5. X-ray Photoelectron Spectroscopy

The XPS measurements were obtained using a Kratos (Manchester, UK) AXIS Supra system. This system is equipped with a 500 mm Rowland circle monochromated Al K-α (1486.6 eV) source and combined hemi-spherical analyzer (HSA) and spherical mirror analyzer (SMA). A spot size of hybrid slot (300–700 µm) was used. All survey scan spectra were collected in the 5–1200 binding energy range in 1 eV steps with a pass energy of 160 eV. High resolution scans of two regions were also conducted using 0.05 eV steps with a pass energy of 20 eV. An accelerating voltage of 15 eV and an emission current of 15 mA were used for the analysis.

#### 2.2.6. X-ray Diffraction

The X-ray diffraction (XRD) patterns of pristine LF and its treated forms were obtained using an Empyrean powder X-ray diffractometer (PANalytical, St-Laurent, QC, Canada). The powdered samples were mounted as methanol films in a horizontal configuration where the XRD patterns were measured in continuous mode over a fixed range of 2θ-values (7–90°) with a scan rate of 3.2° min^−1^. A monochromatic Co−Kα1 radiation was used where the applied voltage and current were set to 40 kV and 45 mA, respectively. The crystallinity index (*C_r_I*) was estimated using a percentile scale by Equation (1), where *I*_22_ is the peak intensity at 2θ near 22°, and *I*_18_ is the amorphous intensity value at 2θ near 18°. The empirical method described by Equation (1) is a well-established approach [24,25]:(1)$$C_{r}I=\frac{I_{22}-I_{18}}{I_{22}}\times 100\%$$

#### 2.2.7. Differential Scanning Calorimetry

The DSC profiles were acquired using a TA Q50 analyzer (TA Instruments, New Castle, DE, USA). The respective samples (~20 mg) were heated from 30 to 150 °C in hermetically sealed pans. The scan rate was set at 10 °C·min^−1^, and nitrogen gas was used to regulate the sample temperature and purging environment.

#### 2.2.8. Gravimetric Water Swelling Tests

The swelling properties of LF and the treated forms (CF and PAF) were evaluated for their uptake of liquid water at equilibrium. The degree of swelling (*S*) in water (*S_W_*) for the samples were calculated using Equation (2), where the sample (~20 mg) was measured using a Mettler Toledo ML204 220 g ± 0.1 mg analytical balance (Cole-Parmer, Vernon Hills, IL, USA) and equilibrated in Millipore water (7 mL) for 48 h:(2)$$S_{w}\left(\%\right)=\frac{W_{s}-W_{d}}{W_{d}}\times 100$$

In Equation (2), *W_s_* refers to the wet sample weight, and *W_d_* refers to the dry sample weight after oven drying at 60 °C to a constant weight.

#### 2.2.9. Water Retention Value (WRV)

A fixed amount of LF, CF, or PAF sample (ca. 40 ± 0.01 mg) was equilibrated in deionized water for 1 h, followed by centrifugation (Precision Micro-Semi Micro Centricone, Precision Scientific Co.) at 4000 rpm to separate the supernatant. The respective sample weight of hydrated (*M*_3_) and oven dry (*M*_4_) samples were recorded in triplicate to a constant weight (±0.01 mg). The samples were dried in a conventional oven at 105 °C, followed by desiccation for 12 h prior to recording the final weight (*M*_4_). The *WRV* (%) was estimated using Equation (3):(3)$$WRV\left(\%\right)=\frac{M_{3}-M_{4}}{M_{3}}\times 100$$

#### 2.2.10. Water Vapour Adsorption Isotherms

The adsorption isotherms were established using the Intelligent Gravimetric Analyzer (IGA) system IGA-002 (Hiden Isochema, Warrington, UK). In this experiment, solid samples (ca. 40 mg) were placed in a stainless steel chamber attached to a microbalance with an uncertainty of less than ±1 µg, giving an uncertainty in moisture content below 0.1%. The sample container was housed in a vessel equipped with a thermostat that allowed for ultra-high vacuum conditions. The desired temperature inside the vessel was controlled accurately by an external water bath. Prior to the start of the isotherm measurements, samples were thoroughly dried at 70 °C in vacuo (≈10^−8^ mbar) for 6 h. The IGA system and analyzer controls the input and output valves to achieve the required relative partial pressure (P/P_0_) in the chamber. After the sample weight reached equilibrium partial pressure within 1.0 mbar (P), the IGA gravimetric analyzer moves to the next incremental P for the isotherm at 25 °C for a range of values using 5 mbar increments from 0 to 30 mbar.

#### 2.2.11. Dye-Probe Kinetic Uptake Studies

The sorption uptake kinetics of the pristine (LF) and treated flax fibers (CF and PAF) were measured using a one pot kinetic method where the adsorption of methylene blue (MB) was monitored as a function of time in one- and two-component dye systems, as described elsewhere [26]. Briefly, ca. 120 mg of the flax fiber sample was added into a dialysis bag that was clipped at both ends, analogous to a tea bag sachet configuration. The adsorbent was immersed in 150 mL MB dye (~10 μmol) solution in Millipore water at ambient pH equipped with a Teflon stirring bar. Aliquots of MB solution (2.6 mL) were taken at variable time intervals (*t*) during continuous stirring at ~150 rpm, where the supernatant dye concentration was measured using a UV-vis spectrophotometer. The dye uptake (Q_t_; mmol/g) was plotted as a function of time (h).

For the batch sorption studies, the flax fiber samples (ca. 10 mg) were added to 4-dram glass vials that contain MB solution (10 mL) with a variable initial concentration (*C_o_*) from 0.2 to 10 mM. The glass vials were sealed with parafilm-lined lid that was subjected to mixing at ca. 150 rpm in a horizontal mechanical shaker for 24 h. The residual concentration (*C_e_*) was measured by UV-vis spectrophotometry, where the removal efficiency (*E_R_*, %) was computed using Equation (4), where *C_o_* is the initial (before adsorption) and *C_e_* is the residual (after adsorption) dye concentration at equilibrium:(4)$$E_{R}\left(\%\right)=\frac{C_{0}-C_{e}}{C_{0}}\times 100$$

## 3. Results

### 3.1. Characterization of Linen Fiber and Treated Forms

The controlled oxidation of flax fibers relied on the use of perchloroacetic acid and chlorite, respectively. In turn, this approach of biomass conversion was used as a strategy to modify the surface chemical properties of LF, where characterization of the structure and the physicochemical properties are outlined below.

#### 3.1.1. Compositional Analysis of Pristine and Treated Linen Fiber

The structural composition of linen fiber (LF) before and after treatment with chlorite (CF) and peroxyacetic acid (PAF) was determined using the Ankom method. The biopolymer content (%) of the pristine and modified fibers is presented in Table 1, where good agreement with the literature values is noted [27,28]. The relative composition of the pristine LF material was altered upon modification, according to the CF and PAF samples. While the compositional changes do not follow a clear trend and a resulting variation in the chemical composition of the fibers was noted. The variation in the hemicellulose, cellulose, lignin and pectin content affects the textural properties of the fibers, in line with the composition analysis in Table 1. The reduction of the pectin fraction in the modified samples may relate to hydrolysis of this polysaccharide upon treatment with chlorite and peroxyacetic acid [29]. The results in Table 1 indicate that the water uptake properties for the various fiber samples (LF, PAF, and CF) may vary according to the relative cellulose, hemicellulose, and pectin content. It is noteworthy that there is an offset of ~5% in the total composition for the LF sample which is accounted for by small fractions (ca. 2–5%) of phenolics, protein residues, and ash content assigned to waxes and other inorganic compounds that may not be reflected by the results of the pristine LF sample in Table 1. The adsorption and water swelling properties, and thermal stability of the biomass fiber composites are presented in the following sections (*vide infra*).

#### 3.1.2. Particle Size Distribution (PSD) and Powder X-ray Diffraction (PXRD)

The PSD and PXRD results in Figure 1a,b for LF and its treated forms provide valuable information regarding the composition and crystallinity of the materials. The PSD of the chemically treated materials (200–300 nm) was lower relative to the pristine LF materials (500 nm). The reduced particle size may relate to the disintegration of the fiber strands and/or loss of fiber constituents upon chemical treatment. Chemical treatment of cellulosic fiber materials can alter the crystalline structure via disruption of H-bonding, causing a material to have an apparently lower PSD [30].

The PXRD and crystallinity data (Figure 1b and inset) further support the changes in the fibril structure of LF upon chemical treatment. Evidence of the reduced crystallinity of the treated materials is shown by the broader features in the XRD patterns, along with the decreased crystallinity index (C.I.; cf. Figure 1b, inset). The observed changes concur with the composition of cellulose/hemicellulose in the treated materials, in agreement with the results in Table 1. Despite the removal of other biopolymers (e.g., hemicellulose, pectins and lignins) upon chemical modification, the estimated values of C.I. (LF>CF>PAF) indicate greater amorphous character of the treated samples, in agreement with the broader XRD line features. The amorphous nature of the treated forms (CF and PAF) may relate to the greater porosity upon removal of pectins, lignins, and hemicellulose. The creation of structural defects via chemical etching results in an apparent increase in the level of amorphous cellulose, in agreement with the results in Table 1. Note also that the treatment of LF using peroxyacetic acid (PAF) did not seem to have much effect on hemicellulose (Table 1) which may contribute to its lower crystallinity, consistent with the CI data above [31,32].

#### 3.1.3. Thermal Gravimetric Analysis (TGA) and FT-IR Spectroscopy

IR spectra and TGA profiles were used to characterize the chemical functionality and thermal stability of the biomaterials, respectively [33,34]. More recently, Karoyo et al. [26] reported the presence of resolved thermal events in the TGA profiles to gain insight on the composition of biopolymer composites. The TGA and FT-IR profiles of pristine LF and its treated forms (CF and PAF) in Figure 2 reveal changes in the structure of linen fibers before and after chemical treatment.

In Figure 2a, the TGA derivative (DTG) plot reveals two bands in the defined temperature region, in agreement with another lignocellulosic fiber study [31]. The weight loss event near 100 °C relates to dehydration from the surface/pore sites of cellulosic fibers [35], where the contribution is relatively low (~1%) for the LF samples due to sample pre-drying. The thermal events ~200–350 °C correspond to the decomposition of the cellulosic substances (mainly cellulose, and hemicellulose/pectin). The degradation of CF (*T*_deg_ ~ 300 °C) and PAF (*T*_deg_ ~ 325 °C) shifts to lower temperatures relative to the pristine LF sample (*T*_deg_ ~ 350 °C). The reduced thermal stability of the chemically treated fibers, especially the CF sample, relates to attenuated hydrogen bonding of the fibril biopolymer components and the partial oxidation of surface functional groups, where reduced adhesion among the biopolymer constituents of the modified fibers vs. pristine LF [34] concurs with the PXRD results. The relative composition of lignin constituents is noted, where a higher decomposition temperature (ca. 200–900 °C), is expected to affect the stability of the samples herein. Among the treated samples, PAF is expected to have greater thermal stability as compared with CF, in agreement with its higher lignin content (cf. Table 1), as evidenced in Figure 2a.

The superimposed weight loss results reveal a greater decrease for LF (ca. 75%) compared to PAF (70%) and CF (ca. 62%), where weight loss values over a similar range is known for pristine and modified cellulosic fibers [31]. The CF sample displayed the least weight loss that concurs with the significant oxidative disintegration of cellulosic/non-cellulosic biopolymer fractions. The chlorite ion is a stronger oxidizing agent under the given conditions and may contribute to further disintegration of biopolymer constituents. The greater weight loss associated with the LF sample implies a greater variation of constituent composition associated with main fiber biopolymers and ash/inorganic components, compared to the treated forms, as described above. Moreover, the trend in relative weight loss among the samples (LF > PAF > CF) may be associated with differences in particle density and heat capacity due to the presence of pores, as supported by the presence of defects in modified fibers in Figure 1b.

Various vibrational bands are noted in Figure 2b; ~3500 cm^−1^ (–OH stretching), ~1735 cm^−1^ (C=O stretching of acetyl groups of hemicellulose/pectin), ~1500–1595 cm^−1^ (C=C in-plane aromatic vibrations due to lignin) and ~1370 and 1430 cm^−1^ (-CH_3_ and C-H symmetric deformation of lignin) [36,37]. Although the FT-IR spectra of the treated LF samples are not markedly different from that of the pristine sample, two distinct features are noted. Firstly, the–OH band (~3500 cm^−1^) for the treated fibers (CF and PAF) has greater intensity and a FWHM that may relate to the oxidation of the reducing end groups of LF (aldehyde to carboxylate) [38]. Similarly, the breakage of inter-/intra-molecular H-bonding within (hemi)cellulose units upon treatment may result in an increase of such bands [39]. Secondly, the decreased intensity of the bands ~1600–1700 cm^−1^ for the treated samples concur with a decrease in the pectin and lignin content (cf. Table 1). The biopolymer content of the flax materials (LF, CF, and PAF) influence the water uptake properties. Hemicellulose and pectins are predominantly amorphous and hydrophilic biopolymers with accessible polar groups that possess high OH-to-C ratios [40]. According to the compositional analysis data in Table 1, the interaction, disintegration or loss of fiber strands and/or biopolymer constituents (such as hemicellulose and pectins) upon chemical treatment lead to lower OH/C ratios, greater fibril defects, PSD and concomitant thermal stability that are consistent with such chemical etching effects. The decrease in the vibrational band intensity ~1595 (lignin) and 1735 cm^−1^ (pectin) further support the complementary results derived from the compositional analysis (cf. Table 1).

### 3.2. Textural and Surface Properties

#### 3.2.1. Scanning (SEM) and Transmission Electron Microscopy (TEM)

The SEM and TEM results for LF and its treated forms (Figure 3) reveal complementary information on the textural properties of the materials. In Figure 3a, the surface of untreated LF appears rough and waxy, as compared to its chemically treated forms (CF and PAF). The latter show quite smooth surfaces with well resolved fibril structure of the (hemi)cellulose constituents. After treatment of LF with chlorite and peroxyacetic acid, greater evidence of micropore features is apparent for CF and PAF samples in Figure 3a, in accordance with fibril defects noted above. The resulting textural porosity relates to the partial removal of constituents in the cell wall biopolymer components such as pectins and lignins, in accordance with the component analysis in Table 1 and the FT-IR results. The TEM results in Figure 3b indicate that the particle size of LF appears similar upon chemical treatment as shown by the comparable particle size for this limited sampling area [41]. The occurrence of fibril etching upon chemical treatment cannot be ruled out based on the TEM results in Figure 3a,b that is also supported by decreased crystallinity results in Figure 1b [39].

#### 3.2.2. Zeta Potential (ξ) Results

The ξ -values for LF and its treated forms were estimated at ambient pH to provide information about the surface charge of the biomaterials. In Figure 4, the chemically treated samples (CF and PAF) have less negative ξ -values (~15 mV), whereas the pristine LF (~35 mV) is ca. two-fold greater. The greater ξ -values for the pristine LF may relate to the greater pectin and lignin content, owing to the presence of various polar functional groups (–COOH, and –OH) and their ionized forms at ambient pH. Treatment of LF with chlorite or peroxyacetic acid results in attenuated OH/C ratios. The diminishing trend in OH/C ratios relate to the removal of hydrolysed biopolymer fractions, fragmentation of certain biopolymer constituents, and/or surface reactions with the chlorite/peroxyacetic acid oxidants [42]. The loss of lignins/pectins and/or oxidation account for the decrease in surface charge observed for CF and PAF [43,44].

#### 3.2.3. X-ray Photoelectron Spectroscopy (XPS)

XPS provides further evidence regarding the surface functionalization of biomass [45], where LF consists of variable levels of hemicellulose, lignin and pectin [46]. The fiber retting process for biomass allows for easy extraction of LF from the plant stem, but has a greater implication on the degradation of lignins, pectins, and hemicellulose fractions over cellulose, due to differences in hydrophilicity [45]. Herein, the chemical treatment of LF resulted in modification of the surface charge and functional groups of the biomass fractions. The XPS survey and spectral deconvolution results for LF and its treated forms are shown in the supporting information (cf. Appendix A) where various regions are displayed for pristine LF and its modified forms: C1s (285 eV), O1s (533 eV) and N1s (398 eV). The deconvoluted spectra show no marked changes in binding energies of LF upon chemical modification (cf. Appendix A). The XPS results are summarized in Table 2 which reveal an increase in the O/C ratios for the LF fiber upon chemical treatment. Chemical modification of LFs may result in the exposure of (hemi)cellulose on the fiber surface due to the loss of surface-bound hydrophobic substances [47,48]. It is important to note the introduction of O-heteroatoms upon oxidation of the flax fibers may likely contribute to higher O/C ratios, which may not be reflected in the estimated results for the ξ -values.

## 4. Discussion

### 4.1. Hydration Properties of Pristine and Treated Linen Fibers

The characterization results outlined above for LF and its treated forms provide insight on the surface, textural and morphological properties of the pristine and treated biomass. With the aid of elemental analysis, microscopy (SEM and TEM), thermal analyses (DSC and TGA), spectroscopy (DLS, FT-IR, XPS, and powder XRD), changes in the physicochemical properties of pristine LF and its chemically modified forms can be evaluated. In particular, alteration and composition of the functional groups result in variable surface charge, porosity, and thermal stability of the LF sample upon chemical conversion to CF or PAF forms. The converted biomass (CF and PAF) is anticipated to have variable hydrophilic character with modified water uptake properties since the affinity of the biomaterials for water depend on various factors: (1) the presence of surface accessible functional groups and surface charge, and (2) the presence of textural porosity (cf. Figure 5).

Adsorption isotherm studies that employ nitrogen gas or isotherms obtained in aqueous media with dye probes such as methylene blue (MB) provide complementary insight on the textural properties (surface area and pore structure) of biomaterials. In Figure 5a–c for LF and its modified forms (CF and PAF), Type II adsorption-desorption isotherms are observed without any evidence of notable hysteresis loops, where a pronounced uptake of N_2_ gas occurs at higher values of P/P_o_ near 0.95. The relative similarity of the features in the adsorption isotherms is evidenced by the comparable textural properties (Table 3) that characterize nonporous materials [23]. One notable difference is the offset for the desorption branch for LF (cf. Figure 5a), where this relates to the greater content of hydrophilic components (pectins/lignins/hemicellulose) over that of the modified fibers (CF, PAF). By contrast, the use of dye adsorption in aqueous media is posited to provide further insight on the textural/surface properties of the LF materials [26]. In Figure 5d, the removal efficiency of MB by LF (E_R_ ~62%) is about 3-fold greater compared to the modified samples (CF and PAF; E_R_ ~21%). The greater adsorptive uptake by LF is consistent with its negative surface charge, in agreement with the presence of hydrophilic biopolymers and the results of Figure 4. By contrast, the uptake of MB is diminished at higher dye concentration in accordance with Figure 5d, where the relative difference in MB uptake by LF and its treated forms (CF and PAF) decrease from ca. 70% to 10% at [MB] = 2 mM versus [MB] = 10 mM, respectively. The latter indicates that pore adsorption occurs at these conditions due to capillary action at elevated dye concentration ([MB] ≥ 1 mM), consistent with the greater chemical potential and driving force at such elevated dye levels. The greater MB adsorption by CF and PAF (relative to LF) at such elevated dye concentration provides support that the treated biomass has greater textural porosity, in agreement with the SEM, kinetic dye uptake results (cf. Figure 5d, inset) and the pore volumes listed in Table 3.

#### 4.1.1. Water Vapour Uptake by Linen Fiber and Its Modified Forms

The water vapour adsorption isotherms at 25 °C for various biomass are shown in Figure 6, where the estimated BET parameters are listed in Table 3. Similar to the nitrogen gas isotherm profiles, the vapour uptake plots for LF and its treated forms adopt Type II profiles [49], where greater water uptake occurred at lower P/P_0_ values near ~0.2. The relative water vapour uptake (*w*/*w* %) for the fiber materials are presented: CF (29%) > LF (24%) > PAF (19%). Similarly, the BET estimates of water vapour adsorption capacity (Q_m_; g_wv_/g_biomass_) are listed: PAF (4.62) > CF (4.45) > LF (4.26). The greater vapour uptake for the modified samples relate to their improved textural porosity and surface-accessibility of the polar functional groups of the biomass upon chemical treatment. In contrast to the N_2_ isotherm results, the uptake for water vapour is more pronounced due to the greater adsorption affinity with the biopolymer surface with H_2_O over N_2_. In particular, the PAF sample was characterized to be more amorphous with a greater hemicellulose fraction, along with greater pore size/surface area over CF (cf. Figure 1 and Table 3). It is important to note that water swelling is more pronounced in the liquid state due to sorption (adsorption + absorption) effects. By comparison, hydration of biomass in the vapour phase adsorption isotherms may contribute to reduced swelling, as compared with sorption in liquid water due to the primary role of surface adsorption contributions in the vapour phase. Greater absorption contributions of the fibers are likely to occur in the liquid phase, in line with the observed swelling effects (cf. *vide infra*; Section 4.1.2).

The textural and surface properties of the various biomass (LF, CF, PAF) in Table 3 were estimated from the nitrogen and water vapour adsorption isotherms. It is noteworthy that the differences in the estimated textural properties from the H_2_O and N_2_ vapour adsorption isotherms relate to differences in molecular properties (e.g., size and polarity). Furthermore, medium effects in liquid H_2_O play a greater role over vapour adsorption, according to the BET textural parameters due to swelling effects for the fiber materials herein. While the adsorption of gases (N_2_ or H_2_O) provide estimates of the textural and surface properties of LF upon chemical modification. The MB dye uptake at equilibrium and kinetic studies reveal that the water (H_2_O*(g)* or liquid *(l)*) uptake by the fiber materials occurs mainly at the surface or within the pore domains. The trends are in accordance with various factors: (1) the molecular size and physical state (*g* or *l*) of the adsorbate, (2) the textural porosity of the fiber, and (3) the availability of surface accessible functional groups. The greater uptake of water vapour by CF and PAF fibers relates to their pronounced textural (and surface) properties. By contrast, pristine LF shows greater uptake in liquid water due to its greater hydrophilic character and the inferred role of capillary-driven sorption.

#### 4.1.2. Gravimetric Water Swelling and Water Retention Value of Linen Fibers

As outlined above, the hydration of biomass depends on the surface and textural properties, along with the solvent molecular properties (size and polarity). Structural variation of the LF material via chemical modification was observed to favour the biomass affinity for water vapour due to the presence of hydrophilic biopolymer components and pore-mediated adsorption processes. By contrast, solvent swelling of biomass provides a measure of the relative role of absorption vs. adsorption processes, along with insight on the relative hydration contributions. The water swelling (S_w_, %) for the biomaterials adopt a trend opposite to that observed for vapour adsorption described above: LF (672) > CF (529) > PAF (447). The reduced swelling of the chemically treated samples may relate to the reduced surface charge, loss of hydrophilic components, and greater textural porosity, in agreement with the recorded ξ-values (Figure 4) and the textural parameters (Table 3). It is noteworthy that the uptake of water vapour and liquid water vary considerably due to contributions that arise from capillary-driven uptake at micropore sites vs. surface-mediated adsorption, as described above. In liquid water, cohesive solvent interactions play a key role in hydration, where SA and electrostatic interactions influence the level of water sorption processes. The propensity of the material to swell in liquid water due to sorption processes afford greater solvent uptake due to allosteric effects which afford an increase in the “apparent” SA of the fiber material. The greater S_w_ values for CF and LF samples relative to PAF agree well with the role of surface charge of the biomass, whereas the former showed greater ξ-values (cf. Figure 4). Further, the variable composition of (hemi)cellulose, pectin and lignin fractions may reveal trends in variable water uptake properties that parallel with that noted for the vapour uptake and water swelling results.

The water retention value (WRV) results provide an indication of the ability of the linen fiber samples to retain water [50]. The disparity between swelling and WRV results in Figure 7 reveals that variable textural properties, surface chemistry, and uptake processes (adsorption vs. absorption) contribute to hydration phenomena to variable extents. The higher WRV for the CF sample may be indicative of its more abundant microporosity and surface charge, in agreement with the SEM and vapour adsorption isotherms according to the greater K_BET_ values (cf. Table 3).

### 4.2. Thermal Analysis and Raman Spectroscopy of Hydrated Biomaterials

Thermoanalytical and spectroscopic methods provide complementary insight on the molecular level processes that accompany biopolymer hydration. In particular, DSC provides insight on the thermodynamic state of water that can be used to study hydration and ascertain the role of chemical treatment on the LF materials [51].

In Figure 8a, the DSC profile depicts the specific thermal responses of LF and its treated forms in their equilibrated and hydrated states as a function of temperature. The DSC profile of the hydrated LF materials are generally broader (100–250 °C), revealing an ensemble of microenvironments at the biomass surface that corroborate the composite microporous nature of these materials. In particular, the thermal transition for the pristine LF is broader and of greater magnitude, as compared with the modified fibers (CF and PAF). The DSC results further reveal the favourable hydration of pristine LF [52], in agreement with the solvent swelling results in Figure 7. The treated samples have reduced water uptake since such oxidative treatment of LF results in partial removal of lignins, hemicelluloses and α-cellulose from the fiber structure, in agreement with the FT-IR and TGA results above [53]. The degradation of hemicellulose is known to occur ~300 °C for cellulosic fibers [54], in agreement with the DSC and TGA results reported herein.

The full-width-half-maximum (FWHM) analysis of the HOD Raman signatures provides further molecular level insight on the fiber hydration properties [55]. The Raman vibrational bands account for changes in polarizability due to interactions that occur between the fibers and bound water. In Figure 8b, two major Raman signatures ~ 2900 cm^−1^ (asymmetric C–H stretching) and 2500 cm^−1^ (OD uncoupled oscillator) are noted for LF and its modified forms in their equilibrium hydrated state. The use of H_2_O/D_2_O mixtures and subsequent analysis of the uncoupled OD oscillator band provide an assessment of the degree of hydrogen bonding, where the accessibility of donor-acceptor groups within the biomass can be evaluated [56,57]. The FWHM values for the OD oscillator bands are inserted within the Raman spectra in Figure 8b (CF > PAF > LF) and reveal variable hydration properties for the pristine and treated LF materials. The greater FWHM results for the CF and PAF further support the abundance of mesopore structure and polar surface sites for these materials and the microporous nature of LFs. As well, the FWHM results account for tightly bound water, in agreement with the WRV values and the isotherm parameters (K_BET_) in Table 3 [58]. The greater swelling for LFs over the modified forms concurs with the more negative ξ –values and broader DSC profiles for such tightly bound water in the pristine fiber materials.

## 5. Conclusions

In this study, pristine linen fibers (LF) were characterized along with their oxidized forms (chlorite- (CF) and peroxyacetic acid- (PAF) treatment). The structural and physicochemical characterization was carried out using several complementary methods: Raman, FT-IR, and XPS spectroscopy, thermal analysis (TGA and DSC), zeta-potential, solvent (water) swelling, and adsorption isotherms with water vapour, nitrogen gas and methylene blue (MB) dye uptake. The results reveal that chemical treatment of LF yields modified fibers (CF and PAF) that display variable morphology, composition and physicochemical (surface and textural) properties. CF and PAF samples showed a reduction in particle size distribution (PSD), lower crystallinity and thermal stability, as compared with pristine LF, as evidenced from XRD, PSD, and thermoanalytical (TGA and DSC) results. These results relate to the potential disintegration of the fiber strands and loss of fiber constituents upon chemical treatment, in agreement with the compositional and XPS results. Nitrogen gas adsorption isotherms were used to estimate the textural and surface properties of the various fiber materials, where the modified samples showed greater textural porosity with limited differences in surface area (SA). Unique solvent swelling of pristine LF in water was observed with a lower water retention value (WRV) when compared with the modified forms, where the swelling of the materials in water (*l*) adopt the following the trend: LF > CF > PAF. By contrast, the adsorption of water vapour by various biomass follow an opposite trend relative to uptake in liquid water, as follows: CF > LF > PAF, due to the dominant role of surface adsorption processes. In liquid water, the trends in hydration of the LF fibers and its treated forms may be favoured via the combined effects of pore- and surface-mediated processes. As such, various factors are likely to play a key role: (1) the molecular size and physical state of the adsorbate (water, *g*/*l*), (2) the textural and surface properties of the fibers, and (3) the presence of accessible surface functional groups for hydrogen bonding or electrostatic interactions. According to the biomass/MB dye adsorption isotherms at equilibrium and kinetic conditions, the uptake of dye by the treated fibers occurred preferentially via pore-mediated processes, in contrast to surface driven processes for pristine LF due to the steric limitations of MB. This study demonstrates a facile tuning of the physicochemical properties of flax biomass via controlled oxidation reactions. An improved understanding is evidenced according to the role of constituent biopolymer fractions in flax fiber biomass. We demonstrate the utility of facile and low-cost chemical methods for biomass transformation to yield biomaterials with variable surface chemical and textural properties. The insights related to *structure-adsorption* properties reported herein contribute to a greater understanding of the hydration properties of the pristine LF and its treated forms. In turn, this study will contribute to the advanced design of sustainable bio-composites for diverse applications such as energy capture, environmental remediation, and biomedical devices.

## Figures and Tables

**Figure 1 bioengineering-07-00038-f001:**
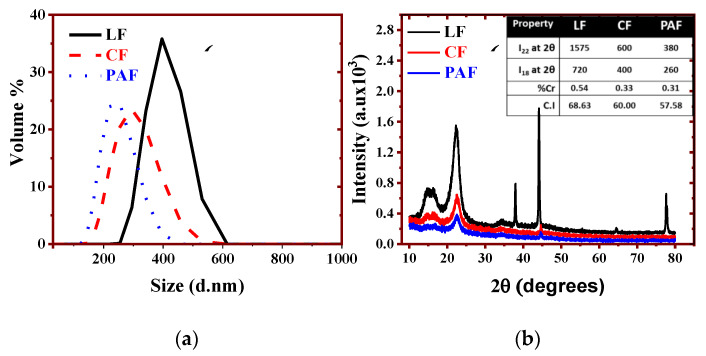
(**a**) Particle size distribution and (**b**) PXRD profiles of linen fiber (LF) and its treated forms (CF and PAF). Figure 1b, inset; XRD parameters for the crystallinity index (*C_r_I*) and peak intensities at 2θ = 22° and 18°.

**Figure 2 bioengineering-07-00038-f002:**
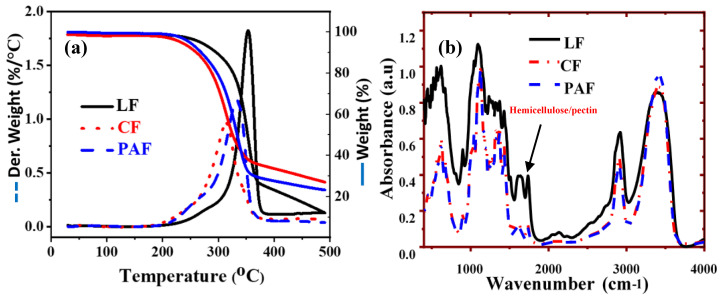
(**a**) Thermogravimetric TG (wt.%) and DTG (%/°C) profiles and (**b**) Fourier-transform infra-red (FT-IR) spectral profiles of LF and its treated forms (CF and PAF).

**Figure 3 bioengineering-07-00038-f003:**
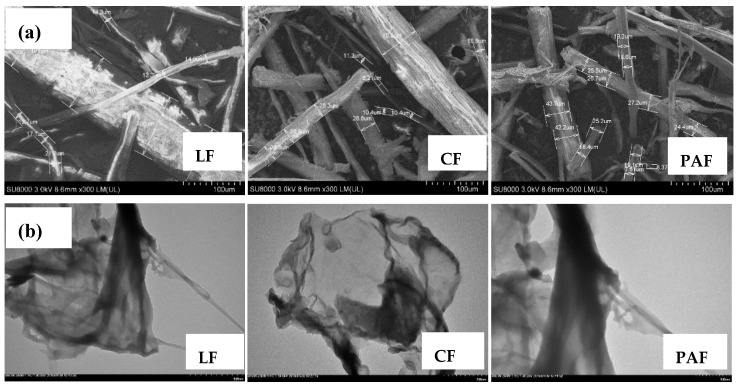
(**a**) SEM and (**b**) TEM images of linen fiber (LF) and its chlorite- (CF) and peroxyacetic acid- treated (PAF) forms acquired at a magnified view (300×).

**Figure 4 bioengineering-07-00038-f004:**
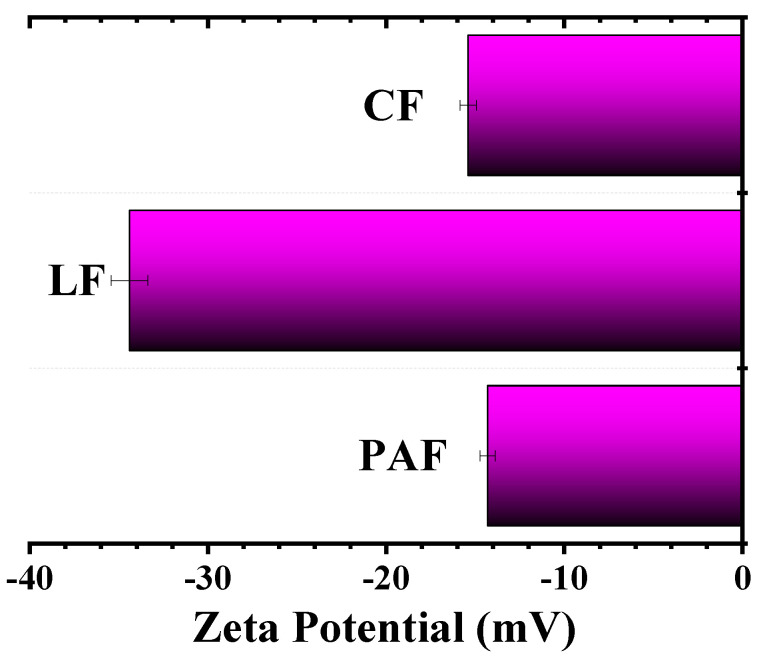
Zeta-potential for pristine linen fiber (LF) and its chemically treated forms (CF and PAF) acquired at pH ~7.

**Figure 5 bioengineering-07-00038-f005:**
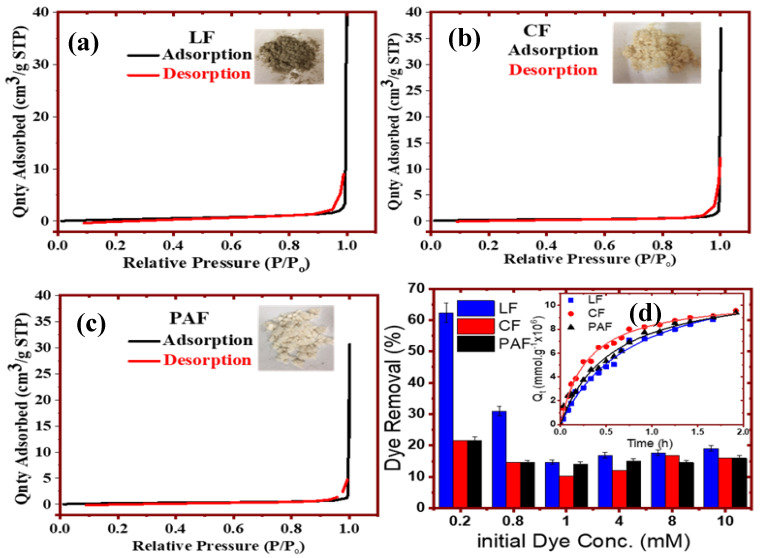
Adsorption properties for linen fiber (LF) materials and its treated forms (CF and PAF): (**a**–**c**) nitrogen gas isotherms, and (**d**) methylene blue (MB) dye removal at equilibrium and kinetic conditions (see inset).

**Figure 6 bioengineering-07-00038-f006:**
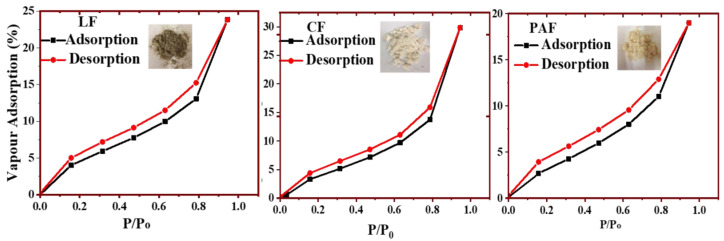
Water vapour adsorption isotherms at 25 °C for linen fiber (LF), and its treated chlorite- (CF) and perchloroacetic acid- (PAF) treated forms.

**Figure 7 bioengineering-07-00038-f007:**
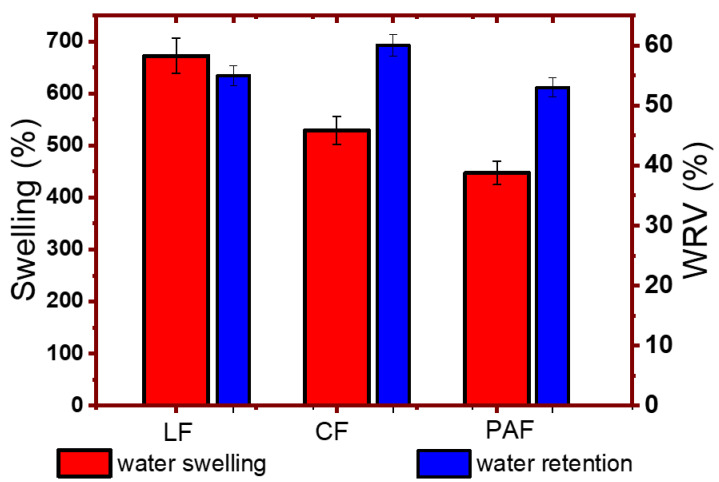
Swelling and water retention values (WRV; %) for linen fiber and its treated forms with an uncertainty of ±3%.

**Figure 8 bioengineering-07-00038-f008:**
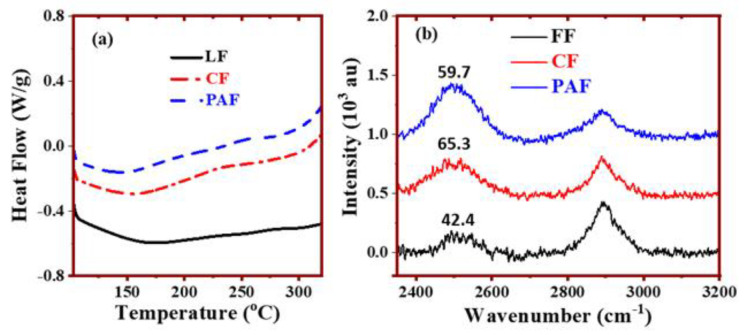
Differential scanning calorimetry (**a**) and Raman spectra (**b**) for Linen Fiber (LF) and its treated forms (CF: chlorite treatment; and PAF: peroxyacetic acid treatment).

**Table 1 bioengineering-07-00038-t001:** Composition (%) of the raw linen fiber with different chemical treatments. (Each assay was run in duplicate and was repeated if the standard error exceeded 3%).

Sample	LF (%)	CF (%)	PAF (%)
Hemicellulose	14.5	11.4	18.9
Cellulose	70.3	85.1	75.7
Lignin	8.6	3.5	5.42
Pectin	2.0	0.0	0.0

**Table 2 bioengineering-07-00038-t002:** Elemental content (wt.%) of Linen Fiber and its modified forms.

Sample Name	FWHM (N1s)	% (C1s)	FWHM (O1s)	% (O1s)	FWHM (N1s)	% (N1s)	O/C Ratio (%)
LF	3.52	79.99	2.56	18.97	3.23	1.03	0.24
CF	3.96	74.23	2.99	24.61	2.75	1.16	0.33
PAF	3.95	76.47	4.00	21.87	3.44	1.66	0.26

**Table 3 bioengineering-07-00038-t003:** Surface and textural properties parameters using BET analysis of the nitrogen adsorption isotherms and water vapour adsorption of linen fiber (LF) and its treated forms (CF, PAF).

Parameters	LF	CF	PAF
*Nitrogen adsorption* (77 K)			
Pore Size (nm)	0.903	1.04	1.08
Surface Area (m^2^/g)	1.29	0.560	0.829
Pore Volume (cm^3^/g)	3.50	7.90	5.00
K_BET_ (L/g)	5.34	7.15	4.80
C	1.15	1.11	1.22
R^2^	0.993	0.996	0.994
Reduced Chi-Sqr	0.392	0.303	0.174
*Water vapour adsorption* (298 K)			
Q_m_ (g/g)	4.26	4.45	4.62
(wt.%)	25	29	21
Surface Area (m^2^/g)	153	159	167

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
