# Peer review of "Flax Biomass Conversion via Controlled Oxidation: Facile Tuning of Physicochemical Properties"

_bioengineering, 2020, doi:10.3390/bioengineering7020038_

Round 1

Reviewer 1 Report

The manuscript “Flax Biomass Conversion via Controlled Oxidation: Facile Tuning of Physicochemical Properties” contains some interesting results. Significant effort was made by authors to characterize the materials prepared using numerous techniques. However, some clarifications should be made when discussing certain results. The following comments might be helpful for authors to improve the manuscript.

  1. Table 1, the total composition for LF sample is 95.4 %. What is the nature of the remaining fraction (4.6 %)?
  2. Section 3.1.3.: the DTG curves show the temperature for mass loss of the samples but not the mass loss percentage. It will be useful for the authors to specify the% mass loss by adding the TG curves.

The degradation of CF and PAF shifts to lower temperatures relative to the pristine LF sample is in agreement with the evolution of particle size distribution, but the inversion of this shift between CF and PAF is not clearly explained.

  1. Lines 328-330: The decrease in the IR band of 3500 cm-1 is attributed to chemical oxidation caused by the chemical treatment of LF. Can the authors give more details on the mechanism of this oxidation?
  2. Line 352: How does Figure 3b (MET results) “indicate that the particle size distribution of LF decreased considerably”; only one particle is shown on each MET image.
  3. Section 4.1.: all the isotherms in the figures 5 (a, b, c) are similar and characteristic of nonporous materials, it may not be necessary to keep all three, one may be sufficient.
  4. Section 4.1.1 & table 3: The specific surface area values calculated from the adsorption of water vapor are too high. Water vapor does not only adsorb physically on the materials but interacts strongly with. In addition, absorption and swelling phenomena are present. The BET method which is based on physical adsorption is not applicable in this case. Subsequently, the real surface area of the samples is that obtained by the adsorption of nitrogen. In my opinion, the textural properties of the samples play certainly a role but this role remains weak compared to the chemical modifications of LF by the chemical treatments.
  5. Line 461: Why does the “capillary action” operate at elevated MB concentration rather than at low concentration knowing that the pores are filled with water in both cases. Also, is this interpretation compatible with the very low porosity of the biomaterials?
  6. Conclusion: The authors recall the results and insist on the modification of LF by chemical treatment. Obviously, the adsorption of MB is not improved by the chemical treatment. What about the potential applications of the chemical treatments studied and/or the chemically modified forms of LF biomass? Can the authors give some details on this subject?

Author Response

Author Response to Reviewer Comments on MS ID:  bioengineering-762300 

Reviewer(s)' Comments to Author:

Reviewer: 1                       

The manuscript “Flax Biomass Conversion via Controlled Oxidation: Facile Tuning of Physicochemical Properties” contains some interesting results. Significant effort was made by authors to characterize the materials prepared using numerous techniques. However, some clarifications should be made when discussing certain results. The following comments might be helpful for authors to improve the manuscript.

  1. Table 1, the total composition for LF sample is 95.4 %. What is the nature of the remaining fraction (4.6 %)?

Response: The authors wish to thank the reviewer for this important comment. It is important to point out that the major biopolymer components of LF have varying compositions with varying ranges, as reported in numerous studies; for example, cellulose (65%–75%), hemicellulose (11%-20%), pectins (1.8%-2.2%), and lignins (1.5%-1.7%) [Bastra, SK, 1998; Lilholt, H. et. al. 1999;  Khalil, A. et al. 2000; Troger, F. et. al. 1998, Dittenber, DB. et al. 2012, etc.). There are, however, smaller fractions related to phenolic compounds, proteins residues and ash consisting of waxes and inorganic compounds that may constitute up to 5 % of LF (Akon, D.E., see the following: http://dx.doi.org/10.5402/2013/186534). The materials in Table 1 were analyzed for the four major components above; whereas, the LF may contain the said residues (phenols, proteins, and ash) in its untreated form, as evidenced by the variable composition of the treated samples. An explanation related to the offset in the composition of LF has been added to the revised MS.  

  1. Section 3.1.3.: the DTG curves show the temperature for mass loss of the samples but not the mass loss percentage. It will be useful for the authors to specify the% mass loss by adding the TG curves.

Response: The authors agree with the reviewer’s recommendation. Therefore, the TG curves were added as recommended, where the mass losses (%) were estimated for LF (~75%), PAF (~70%), and CF (~60%), where the greater weight loss for the LF sample relates to the degradation of main fiber polymer constituents and other non-cellulosic constituents such as ash, proteins and other inorganic components. In contrast, the CF sample exhibited the least weight loss consistent with greater oxidation of polymer constituents in this treated sample. Percent weight loss values in the 60 -70% range and 200 - 350°C region were reported elsewhere for pristine and modified flax fibers (see the following by Cao et al. 2012, Bioresources: doi:10.1016/j.tca.2005.10.016).

The degradation of CF and PAF shifts to lower temperatures relative to the pristine LF sample is in agreement with the evolution of particle size distribution, but the inversion of this shift between CF and PAF is not clearly explained.

Response: The authors acknowledge the reviewer for this insightful comment, where an account of the observed temperature shifts was provided in the revised manuscript. It is understood that, apart from the particle size distribution, the thermal stability of cellulosic fibers and its chemically treated forms relates to the relative composition of the polymer constituents of the samples. For example, samples with a greater % composition of lignin are expected to degrade at much higher temperature consistent with the higher degradation temperature of lignin (~200 – 900 °C), in agreement with a previous report [Ilyas, R.A. et. al. 2017, Bioresources, 12(4)]. This trend noted in the foregoing parallels the results for LF, CF, PAF herein (see Fig. 2a and Table 1). An accompanying explanation has been added to the revised manuscript.

  1. Lines 328-330: The decrease in the IR band of 3500 cm-1 is attributed to chemical oxidation caused by the chemical treatment of LF. Can the authors give more details on the mechanism of this oxidation?

Response: The authors wish to mention here that the original results in Fig. 2b (FT-IR) were not presented in an optimal manner since the baseline for the LF sample was uncorrected. The IR spectrum has been corrected and a revised figure is included in the revised manuscript, to compare with the original data). The revised results show an increase in the –OH band ~3500 cm-1 for the modified materials.   This increase may be related to either oxidation of the reducing end groups of LF (aldehyde to carboxylic groups) [Lazic, B.D. et al. J. Serb. Chem. Soc. 2017] or defects among the intra-/intermolecular hydrogen bonds which tend to increase the number of –OH groups.

  1. Line 352: How does Figure 3b (TEM results) “indicate that the particle size distribution of LF decreased considerably”; only one particle is shown on each TEM image.

The TEM results show that the chemical treatment removes lignin, hemicellulose and it was able to damage cellulose chains which is known to have an effect on the particle size of LF. Also, it is due to removal of the amorphous content of the fiber which results in an increased level of the crystalline cellulose fraction. While there is limited evidence of particle size of the fibers at high magnificiation, it cannot be ruled out that there are changes in the textural porosity of the fiber materials by comparing LF with PAF and CF materials. Further discussionis given in the revised manuscript.

  1. Section 4.1.: all the isotherms in the figures 5 (a, b, c) are similar and characteristic of nonporous materials, it may not be necessary to keep all three, one may be sufficient.

Response: The authors agree that even though the N2 adsorption isotherms are similar, such similarities would not be known before testing. Therefore, the authors feel that it is still valuable to include the isotherms despite the similarities.  By contrast, the desorption curve is characterized by hysteresis loop which is more prominent in the pristine LF compared to the treated forms, and indicate differences in the textural and morphological properties of the various samples. This is in contrast to unmodified LFs of variable particle where the main contribution to nitrogen adsorption was attributed to surface adsorption at the powder grain interface, where negligible adsorption was attributed to adsorption at the pores sites of unmodified LFs (see the following: ACS Omega 2020 5 (11), 6113-6121,   DOI: 10.1021/acsomega.0c00100).

  1. Section 4.1.1 & table 3: The specific surface area values calculated from the adsorption of water vapor are too high. Water vapor does not only adsorb physically on the materials but interacts strongly with. In addition, absorption and swelling phenomena are present. The BET method which is based on physical adsorption is not applicable in this case. Subsequently, the real surface area of the samples is that obtained by the adsorption of nitrogen. In my opinion, the textural properties of the samples play certainly a role but this role remains weak compared to the chemical modifications of LF by the chemical treatments.

Response: The authors agree with the reviewer that the SA estimates obtained using N2 isotherms may offer more consistent estimates due to the absence of strong interactions with the biopolymer constituents of LF biomass.  Nevertheless, the use of water vapour was part of the scope of the current study is anticipated to provide probe dependent estimates the SA, especially for the chemically treated fiber materials. While the difference in SA estimates relate to differences in dipole moment for each backfill gas, the smaller molar volume of water is another important consideration, along with swelling and structural changes in such biopolymer materials (cf. ACS Omega 2018, 3, 11, 15370-1537 https://doi.org/10.1021/acsomega.8b01663). The relative validity of estimates of such textural properties relate to the application in anhydrous (dry) versus wet (hydrated) environments. Additional commentary is provided that address the reviewers concern in the revised manuscript.

  1. Line 461: Why does the “capillary action” operate at elevated MB concentration rather than at low concentration knowing that the pores are filled with water in both cases. Also, is this interpretation compatible with the very low porosity of the biomaterials?

Response: The capillary action is related to adsorption of the dye within the inner pores and it is more relevant for porous material. It is, however, important to note that it is a function of the adsorbate concentration. For example, at lower MB concentration in the current, adsorption of the dye occurs at the surface sites due to their greater accessibility because these materials have low porosity. However, at high MB concentration, adsorption within the pore domains becomes more favorable due to the greater driving force ascribed to the greater chemical potential of the dye at greater MB concentration.

  1. Conclusion: The authors recall the results and insist on the modification of LF by chemical treatment. Obviously, the adsorption of MB is not improved by the chemical treatment. What about the potential applications of the chemical treatments studied and/or the chemically modified forms of LF biomass? Can the authors give some details on this subject?

Response: The authors agree that the conclusions of the paper require improvement in the overall clarity related to chemical modification effects. Herein, it should be noted that the MB was used as a “probe” to study the adsorption properties (surface charge and pore structure) of the various samples. However, given the limited porosity of the materials herein, the MB may not have been the optimal dye probe to assess the textural properties of the samples due to its greater molar volume (and hydrodynamic radius) to that of H2O and N2 in the case of vapour adsorption isotherms. This is evidenced by opposite trends observed for the adsorption of water vapour by the pristine LF and its treated forms (see Table 3). In the latter, the adsorption of the dye via the micropore sites is limited, whereas vapor adsorption seems to involve both surface and pore sites. Furthermore, in the case of MB adsorption, it should be noted that micropore/surface sites undergo some degree of dehydration upon adsorption of the dye. In the case of dye uptake, the treated samples were generally shown to have kinetic rates that exceed or compare to those for untreated LF (see Fig. 5d, inset), and may relate to concentration effects and hydration phenomena, as outlined in the response to comment 7 above. In conclusion, the chemical modification of LF is justified for several reasons; 1) the removal of non-cellulosic components (e.g., lignins and pectins) of LF results in reduced water uptake but may give rise to  more pure samples for application in bio-composite formation; 2) the tuning of the physicochemical properties (e.g. surface charge/hydrophile-lipophile balance, porosity, etc.) results in greater water (vapor) uptake for the treated materials and have potential in energy harvesting in HVAC systems (see the following: DOI: 10.1021/acsomega.0c00762)

In summary, the authors wish to acknowledge Reviewer #1 for the insightful and constructive comments, along with the opportunity to improve the overall quality of this manuscript submission. As well, the manuscript was further edited for language, syntax, and clarity throughout to meet the high standards of his journal.

Reviewer 2 Report

The manuscript by Dehabadi et al. describes preparation and characterization of modified natural fibers. This study covers an interesting subject and fits well current trends regarding use of lignocellulosic materials. The article is well-written, materials and methods are nicely described. However, some questions should be addressed:

General remarks:

Authors should strongly emphasize the novelty of their study. I also recommend more in-depth correlation of results obtained with different methods.

Section 2.1: This part lacks information about drying.

Section 2.2:

Why did Authors decide to measure size with PSD method? In general, this technique is applied for spherical samples.

Results of DSC measurements are affected by sample mass. Why did Authors use such large sample (20 mg)?

Table 1: I guess that reference 27 mentioned in description of the table shouldn’t be there.

Section 3: I find the XRD part of the study a bit confusing. Authors state that CI of samples changed from ca. 68% to ca. 58%. In my opinion, given the patterns shown in manuscript, this difference is more significant. I would recommend re-calculating this parameter using deconvolution of peaks and determining the area of crystalline and amorphous phase. Moreover, one could expect that removal of lignin, pectins and other amorphous parts of fibers (Authors mention it on page 9, line 334) should result in increase of CI. Therefore, Authors are encouraged to discuss theses results with other papers.  

Section 4: “The greater MB adsorption by CF and PAF (relative to LF) at elevated dye concentration” – I cannot agree with this part, differences are not so significant. Absorption properties of produced fibers should also be compared with literature data regarding other lignocellulosic materials.

This section lacks some general summary where all results would be compared and final conclusions would be drawn. At this moment there are some information that seem to be contradictory.

Author Response

Author Response to Reviewer Comments on MS ID:  bioengineering-762300 

Reviewer(s)' Comments to Author:

 Reviewer: 2

The manuscript by Dehabadi et al. describes preparation and characterization of modified natural fibers. This study covers an interesting subject and fits well current trends regarding use of lignocellulosic materials. The article is well-written, materials and methods are nicely described. However, some questions should be addressed:

General remarks:

Authors should strongly emphasize the novelty of their study. I also recommend more in-depth correlation of results obtained with different methods.

The authors wish to thank review for the constructive feedback. The novelty of this study is based on the potential role of tuning the physicochemical and adsorption properties (biomass conversion) of raw linen flax via chemical modification using controlled oxidation processes. The study of the physicochemical properties of LF and its modified forms represent a unique aspect of this study as part of our effort toward understanding of the structure-function properties relevant to adsorption phenomena (further elaborated upon in the response to the specific questions that follow below). In turn, the utility of such modified LF materials via two oxidative methods (chlorite and perchloroacetic acid treatment) are likely to contribute to development of advanced fiber filtration to diverse adsorption-based applications such as environmental remediation, energy harvesting, and carrier systems (see the following: http://dx.doi.org/10.5772/intechopen.69462 and DOI: 10.1021/acsomega.0c00762).  In the latter report by Wahab et al. in ACS Omega (DOI: 10.1021/acsomega.0c00762), the role of surface modification of LFs in the case of such biodesiccants reveal the novelty and key importance of the study reported herein that outlines two facile oxidation processes for modifying the physicochemical properties of flax fibers and it relevance to adsorption-based phenomena.

Section 2.1: This part lacks information about drying.

Response: The authors agree with the reviewer. The information on the drying process was inadvertently left out but has been added to the revised manuscript, where the modified materials were dried at temperature ~60°C after washing with deionized water.

Section 2.2:

Why did Authors decide to measure size with PSD method? In general, this technique is applied for spherical samples.

Response: The average particle size distribution of the flax fiber samples has been studied using Malvern Zetasizer Nano ZS instrument (DLS). DLS is the most used tool for the size characterization of spherical particles but in many cases, it can be used for the size characterization of non-spherical particles in suspension, as reported in other numerous studies (Lotya, M. et al. Nanotechnology 2013, 24(26); Arenas-Guerrero, P. et al. Sci. Rep. 2018, 8)

Results of DSC measurements are affected by sample mass. Why did Authors use such large sample (20 mg)?

Response: The author wish to thank the reviewer for this comment and wish to mention that ranges between 5 - 20 mg have been used for DSC experiments, depending on the density of the subject material. However, smaller masses (1 -3 mg) are recommended for purity determinations. In the present study, the density of the flax materials used allow for ca. 20 mg to be used. This mass was maintained for all samples to ensure consistency and also because that much sample was needed to completely cover the base of the DSC pan with a sample height large enough to ensure uniform heat flow to the sample for good sensitivity.

Table 1: I guess that reference 27 mentioned in description of the table shouldn’t be there.

Response: The authors wish to thank the reviewer for spotting that inadvertent error. The number of reference 27 has been updated to Ref. 26 and removed from Table 1.

Section 3: I find the XRD part of the study a bit confusing. Authors state that CI of samples changed from ca. 68% to ca. 58%. In my opinion, given the patterns shown in manuscript, this difference is more significant. I would recommend re-calculating this parameter using deconvolution of peaks and determining the area of crystalline and amorphous phase. Moreover, one could expect that removal of lignin, pectins and other amorphous parts of fibers (Authors mention it on page 9, line 334) should result in increase of CI. Therefore, Authors are encouraged to discuss theses results with other papers.

Response: The reviewer made a very important observation here, and it is noted that the removal of such components (e.g. lignin and hemicellulose) leads to variation in the crystallinity of such fibers as has reported elsewhere [Ilyas, RA. 2017, Bioresources, 12(4)]. However, the presence of amorphous fractions of cellulose may result in a reverse trend. The authors wish to mention that the empirical formula used here to obtain the crystallinity indices (CI) using XRD peak intensities is a well-established method (Segal, L. et al., DOI. 10.1177/004051755902901003). Further, the use of peak areas will not change the outcome of the CI values in this study as the relative intensities of the various samples vary consistently with the peak areas. Thus, while it was shown that the chemical modification of the flax materials resulted in the removal of the amorphous hemicellulose and lignin components, the reduced CI in the treated materials may relate to the apparent increase in the amorphous cellulose fraction, in agreement with the results in Table 1. Moreover, sharper lines are observed for the XRD patterns of the LF sample relative to the treated forms, indicating greater CI values in the former in agreement with the provided CI data.

Section 4: “The greater MB adsorption by CF and PAF (relative to LF) at elevated dye concentration” – I cannot agree with this part, differences are not so significant. Absorption properties of produced fibers should also be compared with literature data regarding other lignocellulosic materials.

This section lacks some general summary where all results would be compared and final conclusions would be drawn. At this moment there are some information that seem to be contradictory.

Response: The authors agree with the reviewer and the conclusion was edited for clarity that address these concerns in the revised manuscript. However, the authors feel that there is a significant difference in the MB dye adsorption, if such results are presented in a relative sense; e.g. a difference of ~70% vs 10% between LF and the treated forms at [MB] =0.2 mM and 10 mM, respectively. That said, the authors wish to reiterate that the use of MB as a dye probe was mainly intended to probe the surface/textural properties of the LF and its treated forms and to provide further insight on the possible mode of adsorption of water (g/l) via the surface functional groups and/or within pores.  Herein, MB was used as a dye “probe” to study the adsorption properties (surface charge and pore structure) of the various fiber samples. Variations in the estimates of the textural properties of the samples vary due to differences in the molar volume (and hydrodynamic radius) of MB, as compared to H2O and N2 in the case of vapour adsorption isotherms. This is evidenced by opposite trends observed for the adsorption of water vapour by the pristine LF and its treated forms (see Table 3). In the latter, the adsorption of the dye via the micropore sites is limited, whereas vapor adsorption involves both surface and pore sites. Furthermore, in the case of MB adsorption, the treated samples were generally shown to have kinetic rates that exceed or compare to those for untreated LF (see Fig. 5d, inset), and may relate to concentration effects, as noted in the revised manuscript.

In summary, the authors wish to acknowledge Reviewer #1 for the insightful and constructive comments, along with the opportunity to improve the overall quality of this manuscript submission. As well, the manuscript was further edited for language, syntax, and clarity throughout to meet the high standards of his journal.

Round 2

Reviewer 2 Report

Needed explanations were provided, thus I find this manuscript acceptable for publication in Bioengineering.